# Glioblastoma signature in the DNA of blood-derived cells

**Siddharth Jain**[1], **Bijan Mazaheri**[2], **Netanel Raviv**[3], **Jehoshua Bruck**[1]*

**1** Department of Electrical Engineering, California Institute of Technology, Pasadena, CA, United States of America, **2** Department of Computing and Mathematical Sciences, California Institute of Technology, Pasadena, CA, United States of America, **3** Department of Computer Science and Engineering, Washington University in St. Louis, St. Louis, MO, United States of America

* bruck@caltech.edu

## Abstract

Current approach for the detection of cancer is based on identifying genetic mutations typical to tumor cells. This approach is effective only when cancer has already emerged, however, it might be in a stage too advanced for effective treatment. Cancer is caused by the continuous accumulation of mutations; is it possible to measure the time-dependent information of mutation accumulation and predict the emergence of cancer? We hypothesize that the mutation history derived from the tandem repeat regions in blood-derived DNA carries information about the accumulation of the cancer driver mutations in other tissues. To validate our hypothesis, we computed the mutation histories from the tandem repeat regions in blood-derived exomic DNA of 3874 TCGA patients with different cancer types and found a statistically significant signal with specificity ranging from 66% to 93% differentiating Glioblastoma patients from other cancer patients. Our approach and findings offer a new direction for future cancer prediction and early cancer detection based on information derived from blood-derived DNA.

## Introduction

Cancer is the second leading cause of death in the world [1]. It has been widely accepted that cancer is caused by the continuous accumulation of mutations over an individual's lifetime [2]. Most studies in the past have focused on detecting these cancer mutations by studying tumor DNA against normal DNA [3, 4]. This approach has proven useful in identifying cancer genes like TP53, BRCA, HER2, to name a few [3, 5]. Further, these works have also shown that tumor genomes have significantly more genes with repeat instabilities, linking microsatellite instability to colorectal [6] and other cancers [6–11]. Another notable approach that has gained attention in cancer analysis is proposed in [12], where 21 different mutational signatures were identified to characterize different cancer types. Recently, the molecular timing of different driver mutations in the tumor was estimated by analyzing their presence in copied segments spanning the tumor genome [13].

Cancer is a result of the continuous accumulation of mutations. We hypothesize that indications for the mutation activity and likelihood of cancer emergence can be extracted from

**Data Availability Statement:** The BAM files for WXS samples of cancer patients used in the study were obtained from The Cancer Genome Atlas (TCGA). These files have controlled access and cannot be availed publicly. However, request to access TCGA controlled data can be made via dbGap (accession code: phs000178.v1.p1). The

metadata information for the analyzed samples is given in Supplementary CSV Files. The code and necessary documentation for the pipeline used is available at https://github.com/sidjain516/GBM-Classification.

**Funding:** Source of funding: Caltech internal research funding (Mead New Adventures Fund) for Siddharth Jain, Bijan Mazaheri, Netanel Raviv. The funders had no role in study design, data collection and analysis, decision to publish, or preparation of the manuscript.

**Competing interests:** The authors have declared that no competing interests exist.

blood-derived DNA before the formation of tumor cells. In this framework, there are still no cancerous cells, so our approach is to estimate the mutation activity of patients by analyzing blood derived DNA. However, DNA samples represent a snapshot of a current time and we need to estimate the past mutation activity that led to the current samples. How do we extract this hidden mutation history? The idea is to analyze the tandem repeat regions that, metaphorically speaking, are nature's mutation detecting codes [14].

Tandem repeats entail two kinds of events: tandem duplications and point mutations. Tandem duplications involve the consecutive repetition of a substring (e.g. $TC\underline{A}TG \rightarrow TC\underline{A}TC\underline{A}TG$). Point mutations, which include substitutions, insertions, and deletions, are single changes in the DNA (e.g. $AC\underline{T}G \rightarrow AC\underline{A}G$). When these two processes occur in the same location, point mutations can propagate through tandem duplications, leaving a change in the repeated sequence (see Fig 1(a) and Methods). This allows us to construct the most likely history of tandem duplications and point mutations.

An ideal experiment to test our approach would be to derive the mutation history of all tandem repeat regions in blood-derived DNA before an individual gets cancer and predict their chances of getting different cancer types using these inferred histories. However, this

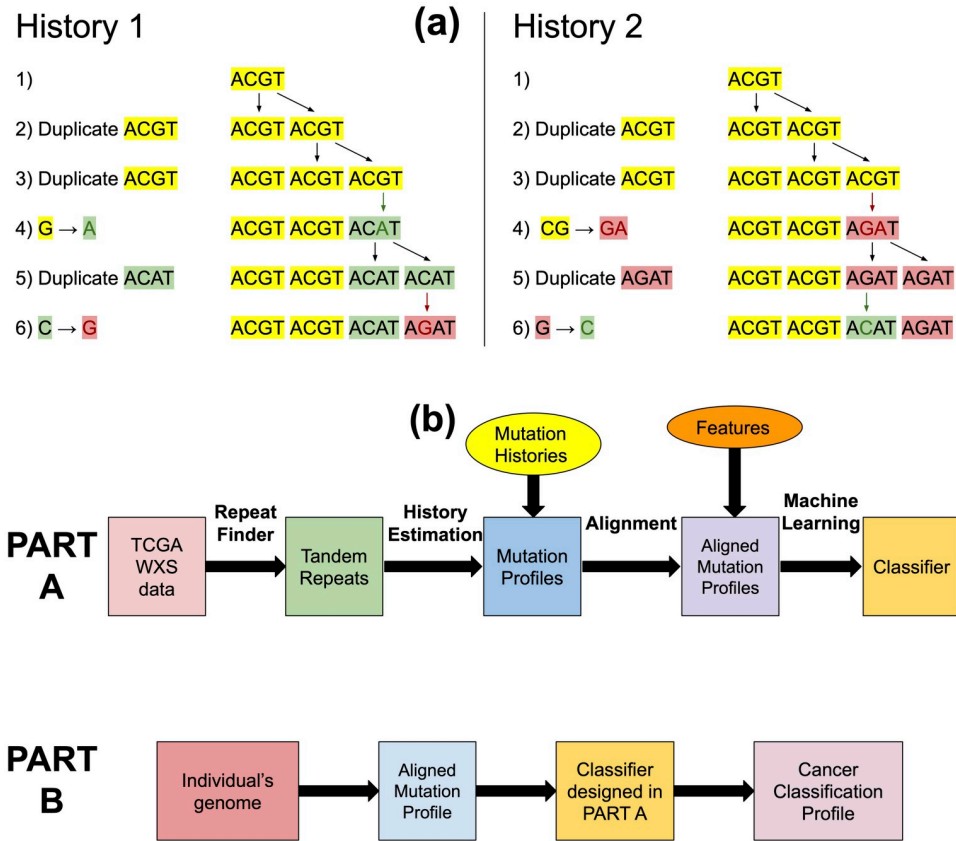

**Fig 1. Pipeline for using mutation history for cancer classification.** (a) Two different mutation histories for the tandem repeat region *ACGTACGTACATAGAT* with pattern length 4 and repeat region length 16. In History 1, only 2 point mutations were needed (marked with green and red respectively). In History 2, 3 point mutations were needed: 1 marked with green and 2 marked with red. In our approach, we would consider History 1 to be more likely as it involves lesser number of point mutations. Therefore, for this tandem repeat region we have that $m = 2$ and $d = 4$. (b) The workflow of our algorithm. In Part A a classifier is trained based on the mutation profiles generated from the blood-derived DNA of cancerous individuals. This part in only performed once per training set. In Part B, the resulting classifier is applied over a given genome to assess an individual's inclination of developing different cancers.

**Table 1. Blood-derived WXS DNA samples.** Number of samples for each cancer type. In total, the number of blood derived WXS DNA samples are 3874. The sample metadata information is provided in the S1–S11 Files.

| Samples | |
| --- | --- |
| Cancer | Blood Derived Normal |
| SKCM | 344 |
| PAAD | 153 |
| STAD | 396 |
| BLCA | 393 |
| PRAD | 440 |
| LGG | 513 |
| LUAD | 411 |
| THCA | 432 |
| LUSC | 316 |
| HNSC | 190 |
| GBM | 255 |

experiment would require DNA samples of cancer patients when they were healthy which turns out to be difficult to obtain. Therefore, we settled for an approximation of this experiment. In this new experiment, we use blood-derived DNA of patients with different cancer types from The Cancer Genome Atlas (TCGA). The blood-derived DNA here serves as a *proxy* for the *past* DNA which the patient had before getting cancer. Now, we search for associations between the mutation histories of tandem repeat regions of blood-derived DNA with the cancer of the individual. We conducted this experiment on 3874 samples obtained using Whole Exome Sequencing (WXS) on TCGA [15] (see Table 1) and found that the mutation histories of tandem repeat regions in blood-derived DNA of patients with Glioblastoma (GBM) is statistically different from those of patients with other cancer types which we describe in detail next.

## Materials and methods

### WXS data

We used exome data from "blood derived normal" samples in the TCGA [15] database, details about which are provided in the S1–S11 Files. The BAM file for each sample was aligned against hg38. All the autosomes from each sample were recovered using samtools [16].

### Algorithms

Our algorithms are partitioned to Part A and Part B (see Fig 1(b)). Part A is only performed once, where Part B is performed whenever cancer prediction is required. In Part A, a dataset of blood-derived DNA is first processed by the Benson [17] and Tang *et al.* [18] algorithms to deduce the mutation profiles. Then, these vectors are aligned by a dynamic programming algorithm to resolve missing regions. Finally, the aligned vectors are fed into a training algorithm to produce a classifier. In Part B, this classifier is applied over any individual's genome, to assess the overall probability to contract any of the cancer in question.

### Tandem repeat detection and mutation history estimation

*Tandem duplications* are consecutively repeated patterns caused by replication slippage events [19, 20], in which a pattern is duplicated next to the original. For example, the following shows two tandem duplications of length 4, where the duplicated part is highlighted in *italics*. The

underlined segment is the *microsatellite* or *repeat region*.

$$\text{ATGAC}GTGA\text{GT} \Rightarrow \text{ATGAC}\underline{GTGAGTGA}\text{GT} \Rightarrow \text{ATGAC}\underline{GTGAGTGAGTGA}\text{GT}. \tag{1}$$

The *pattern* of a region is the short strand which repeats itself. The *copy number d* of a repeat region indicates the number of times that the pattern is repeated. For example, the pattern of the underlined repeat region in the right hand side of (1) is GTGA, and its copy number is 3.

Microsatellites are usually accompanied by various types of errors: substitutions (replacement of one nucleotide by another), deletions (omission of a nucleotide), and insertions (addition of a nucleotide). The total number of substitutions, deletions, and insertions in a repeat region is called the *error number m*. For example, the following shows the contamination of (1) by 1 substitution, 1 deletion, and 1 insertion (highlighted in *italics*).

$$\begin{aligned}
\text{ATGAC}\underline{GTGAGTGAGTGA}\text{GT} \quad &\Rightarrow \text{ATGAC}\underline{GT\textit{T}AGTGAGTGA}\text{GT} \\
&\Rightarrow \text{ATGAC}\underline{GTTAGGAGTGA}\text{GT} \\
&\Rightarrow \text{ATGAC}\underline{GTTAGGAGTGA}\textit{G}\text{GT}.
\end{aligned} \tag{2}$$

Clearly, the copy number of (2) is 3 and its error number is 3, and hence its mutation index is $(m, d) = (3, 3)$. In the first step of Part A we use the Benson Tandem Repeat Finder to detect repeats with consensus pattern size at most 10 and copy number at most 100. These size limitations mean we only consider regions smaller than 1000 nucleotides. The single block version of the duplication history estimation algorithm given in Tang *et al.* [18] was then applied to each tandem repeat region to obtain the respective *mutation index* = $(m, d)$. The aggregation of these $(m, d)$ values gives a vector twice the size of the number of repeat regions, which we call an individual's *mutation profile*. Since, TCGA data is WXS, we only calculated a unique *mutation profile* of an individual's exome.

## Alignment

Following the completion of the Benson and Tang *et al.* algorithms, it was sometimes the case that certain repeat regions appeared in some patients and did not appear in others. In addition, minor differences were observed in the patterns of identical repeat regions in different individuals. As a result, a technical difficulty arose in handling the input to the learning algorithm. Consider the following two patients, in which the repeat regions are underlined.

$$\text{Patient 1}: \underline{\text{AAAAAAA}}\text{CGATCGAGTTCAGTATTGC}\underline{\text{CGCGAGCG}} \overset{\substack{\text{Benson} \\ \text{Tang } \textit{et al.}}}{\Rightarrow} \left(\text{A}: (0,7), \ \text{CG}: (1,4)\right)$$

$$\text{Patient 2}: \underline{\text{AAAAAAAA}}\text{CGA}\underline{\text{CGTACGTACGTA}}\text{TTGC}\underline{\text{CGCGCG}} \overset{\substack{\text{Benson} \\ \text{Tang } \textit{et al.}}}{\Rightarrow} \left(\text{A}: (0,8), \ \text{CGTA}: (0,3), \ \text{CG}: (0,3)\right)$$

The success of machine learning depend on the detection of patterns in specific *positions* of feature vector, so entries which correspond to the same repeat region must also be placed in the same position for all inputs. This is clearly not the case in the above example, in which the second entries of the vectors correspond to different repeat regions.

This issue is resolved by using a dynamic programming alignment algorithm. In this algorithm, a similarity score is computed recursively for each possible alignment, and the alignment which leads to the best possible score is chosen. Each possible alignment is defined as the sum of normalized *edit-distances* that is, the minimal number of insertions, deletions, and

substitutions that are required to transform one pattern to the other, divided by the average length of the sequences between the patterns of all respective pairs. Further, the distance between any pattern and a "missing pattern", denoted by '–' below, is defined as 0.4. Namely, two patterns whose respective normalized edit distance is less than 0.4 were considered to be equal for the sake of the alignment. For example, the vectors above are aligned in the following way.

$$(\texttt{A}:(0,7), \ \texttt{CG}:(1,4)) \qquad \overset{\text{Alignment}}{\Rightarrow} \ (\texttt{A}:(0,7), \qquad - \qquad\qquad , \texttt{CG}:(1,4))$$
$$(3)$$
$$(\texttt{A}:(0,8), \ \texttt{CGTA}:(0,3), \ \texttt{CG}:(1,3)) \qquad \overset{\text{Alignment}}{\Rightarrow} \ (\texttt{A}:(0,8), \ \texttt{CGTA}:(0,3) \quad , \texttt{CG}:(1,3))$$

The score for the alignment (3) is $d_e(\texttt{A}, \texttt{A}) + d_e(-, \texttt{CGTA}) + d_e(\texttt{CG}, \texttt{CG}) = 0 + 0.4 + 0 = 0.4$, where $d_e$ denotes edit distance. For comparison, the alternative alignment

$$(\texttt{A}:(0,7), \ \texttt{CG}:(1,4)) \qquad \overset{\text{Alignment}}{\Rightarrow} \ (\texttt{A}:(0,7), \ \texttt{CG}:(1,4) \qquad , \qquad - \quad )$$
$$(4)$$
$$(\texttt{A}:(0,8), \ \texttt{CGTA}:(0,3), \ \texttt{CG}:(1,3)) \qquad \overset{\text{Alignment}}{\Rightarrow} \ (\texttt{A}:(0,8), \ \texttt{CGTA}:(0,3) \quad , \texttt{CG}:(1,3))$$

has score of $d_e(\texttt{A}, \texttt{A}) + d_e(\texttt{CG}, \texttt{CGTA}) + d_e(-, \texttt{CG}) = 0 + 2/3 + 0.4 \approx 1.06$, and hence (3) is preferred over (4).

The mutation profile of each individual was aligned against the mutation profile of the reference genome (hg38) by using the method that is mentioned above. The repeat regions that were missing in the reference genome were omitted from these aligned mutation profiles. Further, given the aligned mutation profiles, every '–' is replaced by (0, 0). This gave aligned mutation profiles of the same size that can now be used as features for the learning part described next.

## Machine learning

The aligned mutation profiles were used as features for the learning algorithm. Machine learning classifiers for distinguishing cancers were obtained using two approaches:

**Pairwise classifiers.** We trained a binary classifier for *every pair* of types of cancer, generating $\binom{11}{2} = 55$ pairwise classifiers. The accuracy in either of those classifiers is used as a measure for the "uniqueness" of the mutation profiles that cause a certain type of cancer, and can additionally be seen as a distance measure between different types of cancer. In our machine learning pipeline, there are two steps. In the first step, we perform feature extraction using 4-fold cross validation with xgboost [21] algorithm at default parameters with max-depth = 2. We use the top 30 features extracted in the first step to use them as features in the second step to build the pairwise classifier. We use xgboost (max depth = 1) with 4-fold validation to build each of these pairwise classifiers with top 30 identified features.

**Multiclassifier.** We again perform both the steps here, i.e. feature extraction and classifier building. In both the steps, we use xgboost with 'multi:softprob' setting. For the feature extraction step, we use max depth = 2 in xgboost. After identifying the top 30 features in the first step, we use xgboost again with max depth = 1 and the top 30 features to build the multiclassifier. Again 4-fold cross validation was performed in both steps to avoid overfitting.

**Ethics statement.** The ethics approval to the TCGA data was granted by Caltech Institutional Review Board.

# Results

## Tandem repeat regions with different mutation histories in the blood derived DNA of Glioblastoma patients

We reconstruct mutation histories for all the short tandem repeat regions (pattern length ≤10) in exomic DNA derived from blood cells using the algorithm stated in [22]. We calculate these histories from blood-derived DNA for 3874 patients in TCGA covering the following cancer types—GBM (Glioblastoma multiforme), PAAD (Pancreatic Adenocarcinoma), BLCA (Bladder Urothelial Carcinoma), STAD (Stomach adenocarcinoma), SKCM (Skin Cutaneous Melanoma), HNSC (Head and Neck squamous cell carcinoma), LGG (Brain Lower Grade Glioma), PRAD (Prostate adenocarcinoma), LUAD (Lung adenocarcinoma), LUSC (Lung squamous cell carcinoma) and THCA (Thyroid carcinoma) (see Table 1 for number of samples used for each cancer type). In Fig 2, we show that the distribution of these histories (more precisely the number of point mutations $m$ and the number of tandem duplications $d$) are distinguishable for GBM patients in a number of short tandem repeat regions over the whole exome. For example in the tandem repeats in chr7 (82953700:82953740) (Fig 2(b)) and chr9 (137192553:137192600) (Fig 2(c)), we show using violin-plot that the distribution of the number of tandem duplications ($d$) is different for GBM patients when compared with other cancers. In the tandem repeat at chr7 (2513217:2513275) (Fig 2(a)) and chr12 (125025307:125025331) (Fig 2(d)), the distribution of the number of point mutations ($m$) is distinguishable for GBM patients. More such regions are shown in S1–S7 Figs. We built a gradient boosting based classifier [21, 23] for quantifying the differences observed in the mutation histories of different tandem repeat regions shown in Fig 2 (see Methods).

## Statistically distinctive signals for Glioblastoma in the DNA derived from blood

Fig 3 presents the performance of the classifier obtained using 4-fold cross validation (see Methods). We first built pairwise classifiers using the histories of short tandem repeat regions in blood-derived DNA as features between all the pairs of cancers considered (GBM, PAAD, BLCA, STAD, SKCM, HNSC, LGG, PRAD, LUAD, LUSC and THCA). The ROC curves in Fig 3(a)–3(j) show an AUC of 0.85 ± 0.03, 0.69 ± 0.05, 0.84 ± 0.03, 0.75 ± 0.03, 0.87 ± 0.03, 0.81 ± 0.02, 0.79 ± 0.03, 0.87 ± 0.03, 0.81 ± 0.04, 0.82 ± 0.03 when GBM is compared against BLCA, HNSC, LUAD, LUSC, PAAD, PRAD, STAD, SKCM, THCA and LGG respectively using the gradient boosting based classifier. Further, we found that the mean validation accuracy for these pairwise classifiers varies from 69% for HNSC to 88% for PAAD signifying that the blood-derived DNA of GBM patients has a signature embedded in short tandem repeat regions which can be used to distinguish it from BLCA, HNSC, LUAD, LUSC, PAAD, PRAD, STAD, SKCM, THCA and LGG cancer types. The sensitivity/specificity plot further solidifies similar findings showing a mean sensitivity range from 72% for HNSC to 82% for PAAD and a mean specificity range from 66% for HNSC to 93% for PAAD (see S8 Fig).

# Discussion

Using our analysis, we show that we can capture the signature of Glioblastoma using mutation histories of tandem repeat regions in blood-derived DNA. We also notice that access to the tumor genome is *not* required to make this inference. There has been recent interest in designing blood tests for multi-cancer detection in the early stages using cell-free DNA [24]. Our findings offer a new approach for Glioblastoma risk assessment and early detection based on

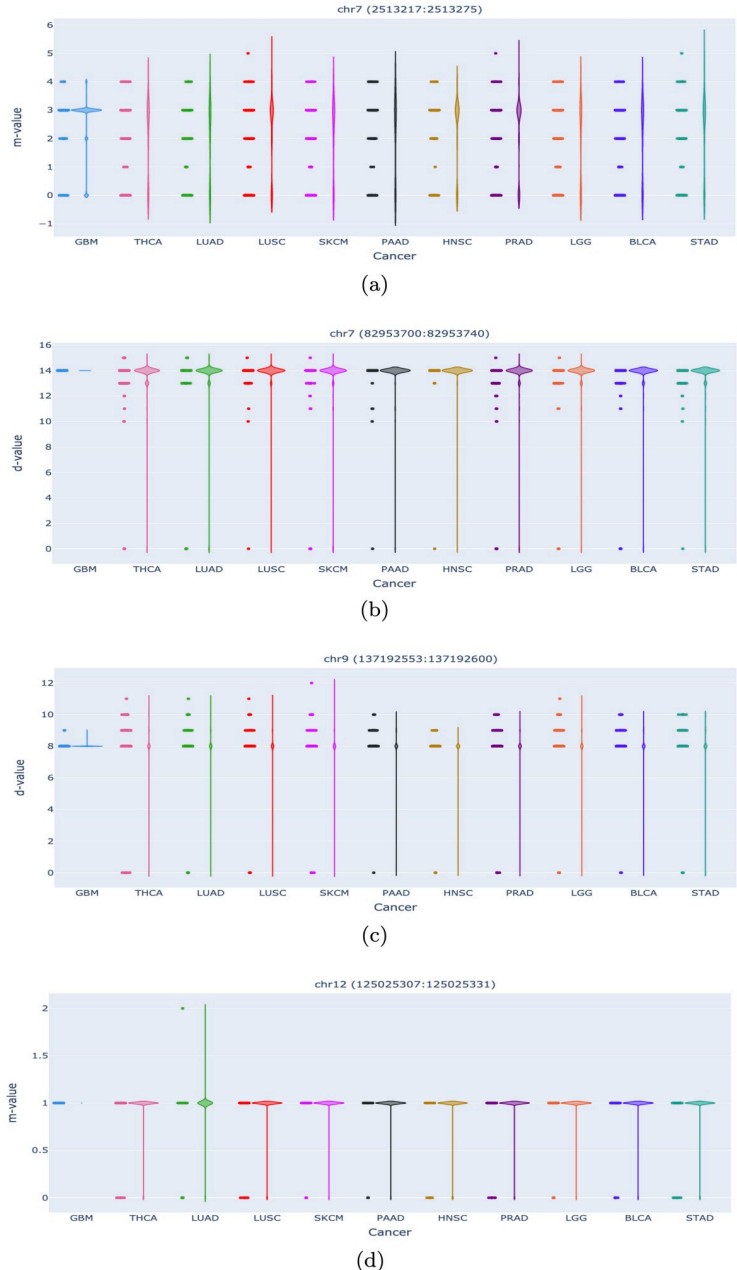

**Fig 2. Distinctive tandem repeat region mutation histories.** Violin plots representing the distribution of $m$ or $d$ in tandem repeat regions in the DNA derived from blood cells for patients with different cancer types. These tandem repeat regions show distinctive distributions of $m$ or $d$ values in GBM patients compared to others leading to a differentiating signature (mutation histories) which is extracted from the DNA of blood-derived cell for GBM patients. An interactive version of these plots is provided at the following links (a), (b), (c), (d).

information derived from the analysis of blood-derived DNA. Further, due to sequencing and TCGA data limitations, our analysis only uses short tandem repeats in the WXS genomic sequences. We believe that if we use (i) WGS data and (ii) account for mutation histories of longer tandem repeats and interspersed repeats, our method might identify signals for other cancer types.

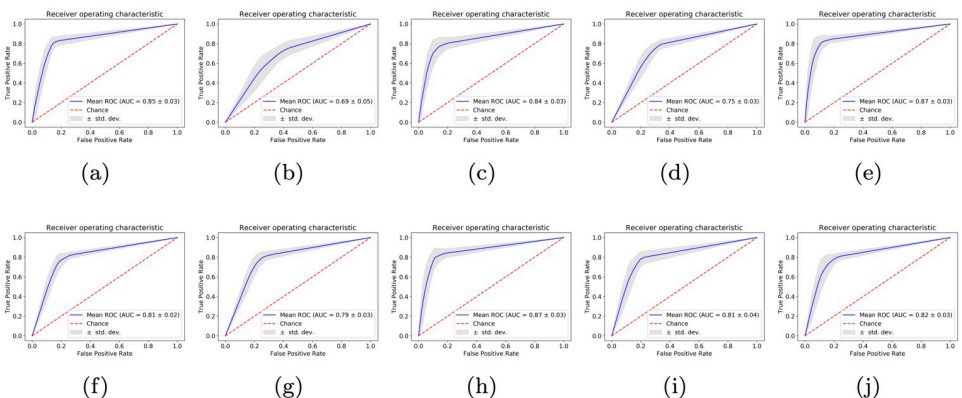

**Fig 3. GBM signature in the DNA of blood-derived cells.** ROC curves for pairwise classifiers built for comparing the mutation profiles of blood-derived DNA of GBM patients against (a) BLCA (b) HNSC (c) LUAD (d) LUSC (e) PAAD (f) PRAD (g) STAD (h) SKCM (i) THCA (j) LGG patients.

Note that, we haven't used blood-derived DNA samples of healthy individuals in the analysis. We didn't include them for two reasons: (a) TCGA dataset is high-quality with 30–40X coverage for WXS samples. We weren't able to find a large dataset of WXS blood-derived DNA of healthy individuals of that quality, (b) By our hypothesis, the blood-derived DNA of healthy individuals may have signals indicative of different cancer types, which could make these samples noisy for our analysis as they may show similar behavior to one or another cancer type(s) used in our study.

Searching for information-containing features within 3 billion nucleotides is a formidable task. This has traditionally been simplified by comparing genomes to extract variants, which compresses the genome into a smaller set of features to analyze. These differences, known as SNPs and CNVs, are central to both Mendelian [25] and Genome Wide Association Studies [3, 4, 26]. However, this form of genome compression loses crucial information regarding how the genome is *changing* by only considering differences in the genome's *current state*. Every individual's genome is controlled by hereditary, environmental and stochastic factors [27]. These factors also vary amongst individuals and can give rise to different risks of disease, but we cannot easily identify these differences from the single-generation SNP and CNV analysis as used in GWAS. Mendelian studies may provide insight into inter-generational processes, but do so at the cost of requiring inter-generational data, which severely limits the scope of a feature search. Even with additional data, Mendelian studies still lack the ability to detect differences in mutation processes that occur throughout one's lifetime.

The use of mutation histories of tandem repeat regions provides a new *computational microscope* to view the genomes. Mutation histories of tandem repeat regions is an *intrinsic* information measure for the genome as the histories are calculated without any comparison with the other genomes. Further, its intrinsic nature allows us to reduce the data-demand as comparisons across different genomes are not needed to extract information. Instead, the tandem repeat regions in a *single* genome provide a glance into its history, capturing information about the individual's mutation dynamics. This ability to reconstruct a genome's history from repeat regions is lost when studies only view differences between individuals. The use of histories expands our access to time-dependent traits which may be essential to understand cancer.

## Supporting information

**S1 Fig. Distinctive tandem repeat region mutation histories in GBM patients.** Violin plots representing the distribution of $m$ or $d$ in tandem repeat areas in the DNA derived from blood cell for patients with different cancer types. These tandem repeat areas show distinctive distribution of $m$ or $d$ values.
(TIF)

**S2 Fig. Distinctive tandem repeat region mutation histories in GBM patients.** Violin plots representing the distribution of $m$ or $d$ in tandem repeat areas in the DNA derived from blood cell for patients with different cancer types. These tandem repeat areas show distinctive distribution of $m$ or $d$ values.
(TIF)

**S3 Fig. Distinctive tandem repeat region mutation histories in GBM patients.** Violin plots representing the distribution of $m$ or $d$ in tandem repeat areas in the DNA derived from blood cell for patients with different cancer types. These tandem repeat areas show distinctive distribution of $m$ or $d$ values.
(TIF)

**S4 Fig. Distinctive tandem repeat region mutation histories in GBM patients.** Violin plots representing the distribution of $m$ or $d$ in tandem repeat areas in the DNA derived from blood cell for patients with different cancer types. These tandem repeat areas show distinctive distribution of $m$ or $d$ values.
(TIF)

**S5 Fig. Distinctive tandem repeat region mutation histories in GBM patients.** Violin plots representing the distribution of $m$ or $d$ in tandem repeat areas in the DNA derived from blood cell for patients with different cancer types. These tandem repeat areas show distinctive distribution of $m$ or $d$ values.
(TIF)

**S6 Fig. Distinctive tandem repeat region mutation histories in GBM patients.** Violin plots representing the distribution of $m$ or $d$ in tandem repeat areas in the DNA derived from blood cell for patients with different cancer types. These tandem repeat areas show distinctive distribution of $m$ or $d$ values.
(TIF)

**S7 Fig. Distinctive tandem repeat region mutation histories in GBM patients.** Violin plots representing the distribution of $m$ or $d$ in tandem repeat areas in the DNA derived from blood cell for patients with different cancer types. These tandem repeat areas show distinctive distribution of $m$ or $d$ values.
(TIF)

**S8 Fig.** Accuracy, Sensitivity and Specificity for Pairwise and Multi Classifiers among different cancer types: Seriation diagram for the pairwise classifiers showing the presence of distinguishing signal between GBM and other cancer types (darker cells) using (a) Mean Validation accuracy and (b) Sensitivity and Specificity. Mean validation accuracy ranges from 69% to 88% when GBM is compared against different cancers in (a). Mean sensitivity ranges from 72% to 82% when GBM is compared against other cancers in (b). Mean specificity ranges from 66% to 93% when GBM is compared against different cancers in (b). A multiclassifier built to compare mutation profiles of GBM patients with other cancer types, here we show that the mutliclassifier

is successful in classifying GBM patients using the Multiclassification probability profile in (c). (TIF)

**S1 File. TCGA-BLCA.** Sample metadata information for TCGA-BLCA samples used in CSV format.
(CSV)

**S2 File. TCGA-GBM.** Sample metadata information for TCGA-GBM samples used in CSV format.
(CSV)

**S3 File. TCGA-HNSC.** Sample metadata information for TCGA-HNSC samples used in CSV format.
(CSV)

**S4 File. TCGA-LGG.** Sample metadata information for TCGA-LGG samples used in CSV format.
(CSV)

**S5 File. TCGA-LUAD.** Sample metadata information for TCGA-LUAD samples used in CSV format.
(CSV)

**S6 File. TCGA-LUSC.** Sample metadata information for TCGA-LUSC samples used in CSV format.
(CSV)

**S7 File. TCGA-PAAD.** Sample metadata information for TCGA-PAAD samples used in CSV format.
(CSV)

**S8 File. TCGA-PRAD.** Sample metadata information for TCGA-PRAD samples used in CSV format.
(CSV)

**S9 File. TCGA-SKCM.** Sample metadata information for TCGA-SKCM samples used in CSV format.
(CSV)

**S10 File. TCGA-STAD.** Sample metadata information for TCGA-STAD samples used in CSV format.
(CSV)

**S11 File. TCGA-THCA.** Sample metadata information for TCGA-THCA samples used in CSV format.
(CSV)

## Acknowledgments

The authors would like to thank Eytan Ruppin for his valuable advice and feedback.

## Data and material availability

The BAM files for WXS samples of cancer patients used in the study were obtained from The Cancer Genome Atlas (TCGA) [15]. These files have controlled access and cannot be availed

publicly. However, request to access TCGA controlled data can be made via dbGap [28] (accession code: phs000178.v1.p1). The metadata information for the analyzed samples is given in S1–S11 Files. The code and necessary documentation for the pipeline used is available at https://github.com/sidjain516/GBM-Classification.

## Author Contributions

**Conceptualization:** Jehoshua Bruck.

**Formal analysis:** Siddharth Jain.

**Funding acquisition:** Jehoshua Bruck.

**Investigation:** Siddharth Jain, Bijan Mazaheri, Jehoshua Bruck.

**Methodology:** Siddharth Jain.

**Software:** Siddharth Jain, Bijan Mazaheri, Netanel Raviv.

**Supervision:** Jehoshua Bruck.

**Validation:** Siddharth Jain, Bijan Mazaheri.

**Writing – original draft:** Siddharth Jain.

**Writing – review & editing:** Siddharth Jain, Bijan Mazaheri, Netanel Raviv, Jehoshua Bruck.

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
