## [Decision Letter · Decision Letter 0]

14 Jul 2021

PONE-D-21-13839

Glioblastoma Signature in the DNA of Healthy Cells

PLOS ONE

Dear Dr. Bruck,

Thank you for submitting your manuscript to PLOS ONE. After careful consideration, we feel that it has merit but does not fully meet PLOS ONE’s publication criteria as it currently stands. Therefore, we invite you to submit a revised version of the manuscript that addresses the points raised during the review process.

We look forward to receiving your revised manuscript.

Kind regards,

Alvaro Galli

Academic Editor

PLOS ONE

Journal Requirements:

3. Please include your tables as part of your main manuscript and remove the individual files. Please note that supplementary tables (should remain/ be uploaded) as separate "supporting information" files.

Reviewers' comments:

Reviewer's Responses to Questions

**Comments to the Author**

1. Is the manuscript technically sound, and do the data support the conclusions?

Reviewer #1: Partly

Reviewer #2: Partly

2. Has the statistical analysis been performed appropriately and rigorously? 

Reviewer #1: I Don't Know

Reviewer #2: Yes

3. Have the authors made all data underlying the findings in their manuscript fully available?

Reviewer #1: No

Reviewer #2: Yes

4. Is the manuscript presented in an intelligible fashion and written in standard English?

Reviewer #1: Yes

Reviewer #2: Yes

5. Review Comments to the Author

Reviewer #1: In this paper the authors describe a potentially useful method and innovative approach to study individual cancer risk.

They describe a computational method to measure the mutational load derived during lifetime of each individuals and the propose to use this methodology as cancer risk predictor.

They declare that is applicable specially or exclusively for Glioblastoma patients.

For this purpose they examine whole exomes data registered in the TCGA for 3874 patients with different type of cancers for mutation histories of tandem repeat regions in “blood-derived healthy DNA”. What does “healthy DNAs” mean?

if it is DNAs coming from lymphocytes of cancer patients, it might be not healthy  at all.

This might invalidate the findings.

So I suggest to send the paper to a journal more related to computational models.

Reviewer #2: The manuscript of Dr. Jain and colleagues is focused on the analysis of the germinal mutational signature in different cancer type patients with the aim of identifying a molecular signature on the DNA can could be informative of which kind of cancer a patient would develop in a tentative predictive approach. Authors, by means of machine learning and computational methods, analyzed the mutational pattern of non-tumor DNA of TGCA-derived genomic data of several cancer types patients. Authors postulated that the historical mutation analysis of that patients could be predictive of kind of cancer. The idea behind the work is very interesting and well inferred in the manuscript. Nevertheless, there are some concerns that need to be fixed by the authors to consider their work for publication in PLOS ONE journal.

Major issues

1. The rationale of limiting mutational analyses of blood-derived DNA to exome is not very clear and should be better explained and discussed, also considering the potential under-represented mutations in non coding regions.

2. How the signature of tandem repeat regions can be different from that of healthy individuals (not affected by cancer of any kind)? Authors did not include in their analysis, nor discussed nothing about. Even if they aligned the genomic information of cancer patients against reference genome hg38, no information is provided, nor discussed, about the possibility that the molecular profiling reported by the authors could discriminate between healthy people from those that will develop cancer (of specific type or of any type).

Minor points

Materials and methods section. “Tandem Repeat Detection and Mutation History Estimation” paragraph. The highlights in bold are not visible in the provided text.

6. PLOS authors have the option to publish the peer review history of their article (what does this mean?). If published, this will include your full peer review and any attached files.

Reviewer #1: No

Reviewer #2: No

---

## [Author Response · Author response to Decision Letter 0]

3 Aug 2021

We thank the reviewers and the editorial board for a careful review of our manuscript. Below, we address the concerns raised:

1. Data availability restrictions: The data is owned by The Cancer Genome Atlas consortium. Since, the DNA data involves human subjects, the data has controlled access and can be accessed by applying using dbGap (https://dbgap.ncbi.nlm.nih.gov/aa/wga.cgi?page=login) under accession code phs000178v1.p1. When we mentioned that the data is available upon request, we meant that the data can be requested using dbGap. Our access to the data also required approval by the Caltech Institutional Review Board. The information about TCGA sample file name and file id used in our analysis is provided in the supplementary CSV files for different cancer types. We have modified the cover letter with these details. 

2. Reviewer 1

- We have removed the use of term healthy from the paper and instead use blood-derived DNA. We have also changed the title of our paper to “Glioblastoma Signature in the DNA of Blood-Derived Cells’’. However, we also want to point out that we carried out the same analysis with DNA derived from normal tissue and obtained similar results. We didn’t include that analysis in the main manuscript since we wanted to point out the existence of indicative cancer signals in blood-derived DNA. Further, our hypothesis is the blood-derived DNA of healthy people may have indicative signals for their future cancer risk. However, this hypothesis can only be validated if access to blood-derived DNA of cancer patients can be obtained when they were healthy.

- We agree with the reviewer that we are proposing a new computational technique to analyze DNA, however in this paper the focus is on the implication of this computational technique on inferring the presence of signal for Glioblastoma in blood-derived DNA. 

3. Reviewer 2

- We would have liked to work with whole genome samples, however TCGA only has Whole Exome samples (WXS). Therefore, we only worked with exomic regions. The method is applicable to whole genome samples and we believe that if one can obtain whole genome samples, the nature of inferred signal for GBM and other cancers might be stronger than what is being reported in the current manuscript. 

- “healthy people DNA”: The reviewer is correct in making this point. There are two reasons, why we didn’t include healthy people in our analysis: 

(i) The WXS data on TCGA is high quality and has a coverage of 30-40X. We couldn’t find a large DNA dataset of healthy people of the aforementioned quality. We looked at 1000 Genome data, however the coverage of that data is very low compared to TCGA data. 

(ii) The second reason is a bit speculative and philosophical. We hypothesize that the blood-derived DNA of a currently healthy individual may have signals for cancer of some type(s) that they can get in the future. If this hypothesis is correct, it may mean, that the DNA of a currently healthy individual may also have signals for cancer(s) they may get in the future. Hence for the purpose of the analysis reported in the manuscript, the individual might fall in a category of some cancer type. 

- We have changed bold font to color red in the new version. 

We have also added some text (highlighted in blue) in the Discussion section summarizing some of the points raised by the reviewers. All the changes made in the manuscript are highlighted in blue.

Thanking you, 

Yours sincerely, 

Siddharth Jain, Bijan Mazaheri, Netanel Raviv, Jehoshua Bruck

---

## [Decision Letter · Decision Letter 1]

17 Aug 2021

Glioblastoma Signature in the DNA of Blood-Derived Cells

PONE-D-21-13839R1

Dear Dr. Bruck,

We’re pleased to inform you that your manuscript has been judged scientifically suitable for publication and will be formally accepted for publication once it meets all outstanding technical requirements.

Kind regards,

Alvaro Galli

Academic Editor

PLOS ONE

Additional Editor Comments (optional):

Reviewers' comments:

Reviewer's Responses to Questions

**Comments to the Author**

1. If the authors have adequately addressed your comments raised in a previous round of review and you feel that this manuscript is now acceptable for publication, you may indicate that here to bypass the “Comments to the Author” section, enter your conflict of interest statement in the “Confidential to Editor” section, and submit your "Accept" recommendation.

Reviewer #2: All comments have been addressed

2. Is the manuscript technically sound, and do the data support the conclusions?

Reviewer #2: Yes

3. Has the statistical analysis been performed appropriately and rigorously? 

Reviewer #2: Yes

4. Have the authors made all data underlying the findings in their manuscript fully available?

Reviewer #2: Yes

5. Is the manuscript presented in an intelligible fashion and written in standard English?

Reviewer #2: Yes

6. Review Comments to the Author

Reviewer #2: (No Response)

7. PLOS authors have the option to publish the peer review history of their article (what does this mean?). If published, this will include your full peer review and any attached files.

Reviewer #2: **Yes: **Roberto Giovannoni

---

## [Editor Report · Acceptance letter]

20 Aug 2021

PONE-D-21-13839R1 

Glioblastoma Signature in the DNA of Blood-Derived Cells 

Dear Dr. Bruck:

I'm pleased to inform you that your manuscript has been deemed suitable for publication in PLOS ONE. Congratulations! Your manuscript is now with our production department. 

Kind regards, 

on behalf of

Dr. Alvaro Galli 

Academic Editor

PLOS ONE